# Spectrum Intelligence via Multi-Agent Hybrid Retrieval-Augmented Generation

## Abstract

Spectrum policy making requires reviewing large volumes of information from different sources, including technical reports, stakeholder filings and incumbent licenses, which makes the process slow and resource-intensive. To speed this process, we propose a Multi-Agent Hybrid Retrieval-Augmented Generation (RAG) system for analyzing spectrum proceeding documents and license data. Through customized prompts and tailored tools, the system demonstrates the ability to handle complex queries and perform multi-step retrieval and analysis. For evaluation, we built a benchmark dataset in the form of question-answer pairs and used it to compare our system with others. Extensive experimental results show that our Multi-Agent Hybrid RAG system significantly improves the understanding of spectrum data, providing more reliable and verifiable support for spectrum policy analysis. Our approach provides new insights into how agentic RAG can leverage heterogeneous structured and unstructured data sources to accomplish complex policy analysis tasks, highlighting its potential for broader applications in domain-specific decision support.

## 1 Introduction

The wireless spectrum is running out as new services and devices compete for limited bandwidth. Therefore, spectrum allocation has become increasingly important. Market-based solutions such as auctions have been tried (National Telecommunications and Information Administration, 2024), but allocation remains largely top-down and policy-driven. Well-designed spectrum policies can improve efficiency and create room for new technologies to grow.

However, spectrum policymaking is not straightforward. It typically requires multiple rounds of discussion among three main stakeholders: the national regulatory authority (NRA, represented in the United States by the Federal Communications Commission (FCC) and the National Telecommunications and Information Administration (NTIA)), incumbent spectrum holders, and entrants seeking access (Mustonen et al., 2017). The NRA usually leads the proceedings and issues the final rules, while incumbents and entrants contribute opinions and technical details. This process generates a large volume of documents with technical and legal language. The NRA must review these materials to understand relevant positions and make informed decisions. Such review is necessary but resource-intensive, slowing regulatory operations (Lu et al., 2025).

The idea of using machines to interpret spectrum policy documents is not new (Berlemann et al., 2005). Recent advances in artificial intelligence (AI), particularly large language models (LLMs) and retrieval-augmented generation (RAG), provide new tools to accelerate this process. LLMs have shown strong ability to understand domain-specific language (Bariah et al., 2023; Min et al., 2025; Oche et al., 2025), and RAG further extends their usefulness by retrieving relevant content from large databases to ground answers in real data (Gao et al., 2023; Rutagemwa et al., 2024; Bornea et al., 2024). This helps address issues such as limited context windows and hallucinations. Agentic RAG goes further by adding planning and reasoning capabilities, enabling multi-step retrieval and multi-document tasks (Singh et al., 2025; Khanda, 2024; Barron et al., 2025).

In this paper, we propose a multi-agent hybrid RAG system designed to support the analysis of spectrum proceeding documents and licensed data. The system enables tasks such as incumbent investigation and opinion collection. We adopt the RAG technology for two key reasons. First,

policy analysis demands high accuracy and transparency. Answers must be grounded in source documents and users can verify them when needed. Second, the relevant data is distributed across multiple FCC and NTIA databases and comes in varied formats. This requires customized data processing and retrieval methods beyond the capabilities of standard language models or web search tools. Specifically, our contributions can be summarized as follows:

1. **Reusable and extensible domain-specific databases :** We systematically extract and process unstructured PDF files containing both license data and policy documents. To capture their distinct structural properties, we build a relational SQL database for license data and a graph-based corpus for policy files. To the best of our knowledge, this is the first effort to transform raw spectrum-related regulatory and licensing documents into structured, machine-readable databases that are directly usable for retrieval. Importantly, our databases are designed to be reusable and easily extensiblenewly released documents can be seamlessly integrated, ensuring long-term community value.

2. **An expert-validated benchmark dataset:** We craft an expert-validated benchmark dataset for spectrum-related question answering, with input from domain experts and PhD researchers to ensure coverage and credibility. The benchmark spans diverse query types, including fine-grained factoid questions, summarization-oriented questions, as well as questions requiring the integration of multiple heterogeneous data sources. While modest in scale, it provides the first systematic benchmark for spectrum-focused RAG.

3. **A multi-agent hybrid RAG system :** We propose a novel spectrum multi-agent hybrid RAG system that leverages a specialized multi-agent architecture to retrieve and reason over heterogeneous sources (tabular license data and policy graphs). This design enables more accurate and comprehensive responses than traditional RAG approaches. Experimental results on our evaluation dataset demonstrate substantial improvements, with our system achieving an overall win rate exceeding 85% against strong baselines.

## 2 RELATED WORKS

**Retrieval Augmented Generation** Retrieval-Augmented Generation (RAG) has emerged as a pivotal paradigm that augments large language models with external non-parametric memory, allowing dynamic fetching of relevant knowledge to overcome the limits of static pretrained models (Min et al., 2021; Gao et al., 2023; Guu et al., 2020). This approach has significantly improved performance on knowledge-intensive tasks by grounding generation in up-to-date evidence, thereby enhancing factual accuracy and reducing hallucinations (Petroni et al., 2021). However, while traditional RAG systems excel with unstructured text, they often fall short in handling numerical or structured data (e.g., spreadsheets or knowledge base tables) (Wallace et al., 2019; Spithourakis & Riedel, 2018; Sundararaman et al., 2020). Standard dense retrievers struggle with precise numeric matches or table schema, limiting the systems ability to retrieve and reason over such information. Recent efforts such as TABLERAG have begun to address this gap by tailoring retrieval and reasoning mechanisms for structured data sources, providing initial solutions for integrating tables and databases into the RAG framework (Chen et al., 2024b).

**Large language model (LLM) agent** Large language model (LLM) agents combine pretrained language models with external tools and structured reasoning to solve complex tasks (Li et al., 2025). They typically operate in a thinkactobserve loop, where the model generates chain-of-thought reasoning to plan actions, invokes external tools or APIs, and incorporates feedback into subsequent steps (Peng et al., 2025; Kim et al., 2025; Fan et al., 2024; Yu et al., 2025). Recent work highlights their effectiveness: ReAct integrates reasoning with tool calls to reduce hallucinations and improve reliability (Yao et al., 2023); Toolformer trains models to autonomously decide when and how to use APIs (Schick et al., 2023). These studies show that LLM agents provide a powerful and general framework for tool use, multi-step reasoning, and adaptive decision-making.

**GraphRAG** GRAPHRAG (Edge et al., 2025) constructs a knowledge graph corpus by prompting LLMs to extract entities and relations from documents, treating entities as nodes and relations as edges. Through this graph structure, GRAPHRAG retrieves entities and relations most relevant to the query, enabling more accurate capture of relationships and better general understanding across

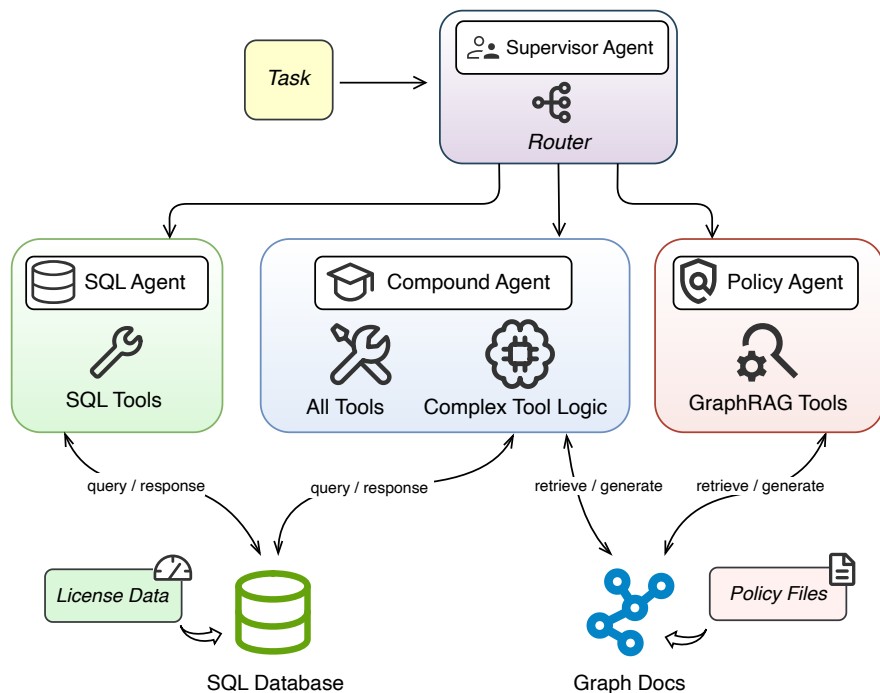

Figure 1: Illustration of Spectrum Multi-Agent Hybrid RAG System

documents, thereby supporting more abstract and general questions than standard RAG. Such a structure is particularly suitable for our policy files, as they mainly consist of interconnected opinions and comments from individuals and organizations.

## 3 METHODOLOGY

### 3.1 OVERVIEW

As illustrated in Figure 1, our proposed spectrum multi-agent RAG system adopts a hybrid retrieval-augmented generation architecture that integrates structured and unstructured spectrum knowledge sources. A Supervisor Agent serves as the router, dynamically dispatching each incoming task to one of three downstream agents: (i) the SQL Agent, which queries structured license data stored in a SQL database using SQL tools; (ii) the Policy Agent, which applies GraphRAG tools to retrieve and synthesize insights from unstructured policy files; and (iii) the Compound Agent, which coordinates complex tool logic by combining both SQL and GraphRAG functionalities when tasks require joint reasoning over structured and unstructured information. This hybrid design enables flexible query routing and domain-specialized processing, thereby ensuring accurate and context-aware responses for a wide range of spectrum-related tasks.

### 3.2 DATA SOURCES

When formulating new spectrum policies or revising existing allocations, two key aspects must be considered: (i) which entities currently hold allocation authority, and (ii) how the public, including relevant stakeholders, view the proposed changes. To capture these dimensions, we collected data from both sources.

Firstly, spectrum licenses can provide information on current spectrum incumbent users. In our dataset, we collected 1,346 licenses[1] across three bands: 12.7-13.25 GHz, 92-94 GHz, and 94.2-95

---

[1]https://www.fcc.gov/licensing-databases/licensing

GHz. The raw data appears as tabular information embedded in unstructured PDF files. Secondly, documents from public consultation process, typically conducted by the FCC and NTIA before policy is adopted, can provide insight into societal and stakeholder perspectives. From this source, we collected 698 documents across seven proceedings[2], totaling 258.5 MB of data. These raw documents are provided in PDF format with text-based content.

### 3.3 DATA PROCESSING

#### 3.3.1 LICENSE DATA PROCESSING

We extract license data from unstructured PDF tables and convert them into structured JSON, where each record contains various fields. Some are strings, such as *service type*, *licensee*, and *transmit antenna*, while others are numerical, including *license order*, *receive latitude/longitude*, and *frequency range*. A well-known limitation of RAG is its difficulty with numerical data (Wallace et al., 2019; Spithourakis & Riedel, 2018; Sundararaman et al., 2020). Trained primarily in semantic space, LLMs often fail with exact numbersfor example, misjudging that 9.11 is larger than 9.2 or producing inconsistent calculations. Applied directly to license data, conventional RAG performs poorly. Queries such as *"Which licenses transmit longitude is 118.838583?"* demand precise numeric matching, but embedding-based retrieval lacks the required accuracy, highlighting the necessity of database structures capable of supporting exact numeric reasoning.

To address this issue, we turn to SQL databases. SQL provides explicit schema definitions, strong typing for both textual and numerical fields, and efficient mechanisms for structured queries with exact matching or range conditions (Mohammadjafari et al., 2025; Hong et al., 2025; Sun et al., 2024). These characteristics make it especially well-suited for handling license data, where questions often involve precise filtering conditions. By storing license records in SQL and equipping the LLM with a set of SQL tools, the model can translate a natural-language query into an appropriate SQL statement (Sun et al., 2024), execute it against the database, and use the retrieved results to formulate accurate answers.

In designing our SQL database, we account for the fact that license records differ substantially across service types. Each service emphasizes different technical parameters. For instance, the metric *EIRP (Equivalent Isotropically Radiated Power)* frequently appears in satellite communication, microwave links, cellular base stations, and Wi-Fi systems. It is more common in high-frequency bands, though it may also appear in lower-frequency services where it is not usually the primary regulatory constraint. Given these variations, we organize the license data into separate tables by service type. Licenses within the same service typically share a similar set of attributes, which allows for cleaner table design and more consistent SQL queries. This table-per-service approach also enhances extensibility. When a new service type is introduced, we only need to create a new corresponding table without modifying the overall database schema. This modularity ensures scalability of the system and prevents disruptive structural changes to the existing SQL database.

#### 3.3.2 POLICY FILES PROCESSING

To process the policy files, we adopt the GraphRAG (Edge et al., 2025) framework to construct a graph-based corpus database. Specifically, each policy document is first segmented into fixed-length text chunks. An LLM is then prompted to extract salient *entities* and their *relationships* from these chunks. The extracted entities serve as graph nodes, and the relationships are represented as edges, thereby forming a knowledge graph. To capture higher-level structures, we apply the Leiden community detection algorithm (Traag et al., 2019) to partition the knowledge graph into hierarchical communities, where each community contains a set of closely related nodes. For each detected community, the LLM is further prompted to generate a report-style summary, which is stored as part of the index and later supports retrieval in the RAG process. This procedure transforms unstructured policy texts into a structured, multi-level graph corpus that facilitates both entity-level retrieval and community-level contextual reasoning.

In building our graph corpus, each proceeding is treated separately since a stakeholder may take different or even opposing positions across proceedings. For instance, NASA may support non-federal usage in one band while opposing it in another. Constructing a separate graph corpus for each

---

[2]https://www.fcc.gov/ecfs/search/docket-detail

Table 1: graph corpora construction results

| Proceeding | Number of Nodes | | | | |
|---|---|---|---|---|---|
| | Level 0 | Level 1 | Level 2 | Level 3 | Level 4 |
| FCC 19 - 38 | 477 | 445 | 133 | 57 | - |
| FCC 24 - 72 | 375 | 329 | 163 | 23 | - |
| FCC 25 - 59 | 1045 | 1008 | 419 | 40 | - |
| FCC 22 - 352 | 2094 | 2051 | 1599 | 571 | - |
| FCC 23 - 158 | 902 | 868 | 405 | 53 | - |
| FCC 23 - 232 | 871 | 794 | 338 | 82 | 58 |
| NTIA National Spectrum Strategy | 4079 | 3960 | 3074 | 679 | 345 |

proceeding prevents such contradictions from being conflated within a single graph and improves scalability: new proceedings can be indexed independently without altering existing corpora.

By default, we employ `gpt-4o` for constructing the graph corpus. We set the text chunk length to 1200 tokens with an overlap of 100 tokens between adjacent chunks, which helps preserve context continuity across segments and ensures that entities spanning chunk boundaries are not missed. In total, we build graph corpora for seven proceedings. Table 1 summarizes the detailed construction results, reporting the number of nodes extracted at each hierarchical level (from Level 0 to Level 4).

### 3.4 SUPERVISOR AGENT

The Supervisor Agent acts as the entry point and decision maker in our system, responsible for routing user queries to the most appropriate downstream agent. To perform this task, we employ an LLM that is carefully prompted with detailed instructions. The prompt includes both an explicit description of the overall task setting and concise explanations of the two databases used in our framework. This enriched context allows the LLM to make more reliable routing decisions by understanding not only the intent of the query but also the available knowledge sources. The full prompt template is provided in Appendix D.1.

Although one straightforward approach would be to route all queries directly to the Compound Agent, we intentionally avoid this design. Instead, we follow the principle that specialized agents should handle specialized problems. When a query can be fully addressed by a lightweight agent (such as the SQL-based License Agent or the GraphRAG-based Policy Agent), there is no need to invoke the more complex Compound Agent. This selective routing strategy (i) reduces token consumption and operational costs by avoiding unnecessary complexity, and (ii) improves system efficiency and scalability. For future extensions, when new task types arise, the framework can be expanded by simply adding a new specialized agent and updating the Supervisor Agents routing prompt to recognize the new task. If the task requires orchestration across multiple tools, the Compound Agent can be extended with the corresponding logic. Crucially, existing unrelated specialist agents can remain unchanged, preserving modularity and minimizing integration overhead.

### 3.5 SQL AGENT

The SQL Agent is designed to handle numeric and table-centric queries by interacting with our relational database. It is powered by an LLM that is prompted with detailed tool-use instructions and error-handling strategies. In particular, we designed four SQL tools : (i) a tool for listing all tables in the database, (ii) a tool for displaying the schema and representative examples of a specific table, (iii) a tool for validating SQL queries, and (iv) a tool for executing SQL statements. The detailed design of these tools is presented in Table 5.

In the prompt design, we explicitly list the usage logic of these tools and guide the LLM to incorporate the outputs of one tool as the basis for the next step in its reasoning process. The detailed prompt template is shown in Appendix D.2. To illustrate how the SQL Agent operates in practice, we further provide an execution case study in Appendix C.1, which demonstrates step-by-step how the agent leverages these tools to retrieve the necessary information for answering a query.

## 3.6 POLICY AGENT

As described in Section 3.3.2, we construct a graph corpus for each proceeding individually. Therefore, before executing any GraphRAG method, the system must first determine which proceeding is most relevant to the users query. To accomplish this, we employ an LLM that compares the topic content of each proceeding with the users question and selects the most appropriate proceeding. This selection function is embedded in every tool of the Policy Agent, making its execution mandatory before any subsequent retrieval is performed. The detailed prompt template, along with descriptions of each proceeding, are provided in Appendix D.3 and Appendix D.4. Once the relevant proceeding has been selected, the next step is to decide which GraphRAG method should be applied. We consider three methods that are particularly suited to our tasks: basic, local, and global.

The basic method is functionally equivalent to naive RAG: it performs semantic matching between the users query and raw text chunks from the corpus, without leveraging the graph structure. The local method searches over extracted entities and relationships, making it more effective for fine-grained questions (e.g., an individuals stance on a specific proposal). The global method instead operates on community reports generated from graph communities, making it more suitable for broader, summarization-style questions (e.g., the overarching themes of a proceeding). Based on the three methods of GraphRAG, we design three hierarchical GraphRAG tools. The detailed tool usage design is presented in Table 5.

In the prompt design, the LLM is guided to prioritize the local or global GraphRAG tools, depending on the granularity of the users query, so as to leverage the advantages of graph-structured retrieval. However, the GraphRAG pipeline may still miss certain entities during graph construction. In such cases, we design to let the basic method provide a fallback mechanism to recover relevant information directly from the raw corpus. Only when local or global retrieval fails is the basic tool invoked, and its output is returned regardless of success. The full prompt design is shown in Appendix D.5.

## 3.7 COMPOUND AGENT

The Compound Agent is designed to handle complex queries that require integrating evidence from both the license database and the policy document corpus. A representative example is: *"As incumbent users in the 12.7–13.25 GHz band, what are their views on sharing this spectrum with terrestrial services?"* Answering such a query involves a two-stage reasoning process: first, the system must query the SQL database to identify all incumbent users operating in the specified frequency range; second, it must retrieve relevant comments or positions of those users from the policy graph corpus. To enable this functionality, the Compound Agent incorporates all four SQL tools from the SQL Agent together with the three GraphRAG tools from the Policy Agent. This integration allows the agent to flexibly call across both domains, chaining SQL-based data retrieval with graph-based policy analysis when necessary.

In order to support effective decision making, we carefully design a compound prompt that (i) provides precise descriptions of both the SQL database and the graph corpora, (ii) specifies when and how to combine SQL and GraphRAG tool calls, and (iii) guides the LLM in handling error cases. To further improve reliability and interpretability, the prompt also incorporates **in-context examples** that demonstrate typical tool-use trajectories, helping the model better understand how to combine and sequence the available tools. The complete prompt specification for the Compound Agent is presented in Appendix D.6. The Compound Agent bridges structured license data and unstructured policy texts, enabling queries beyond the scope of any single specialized agent.

## 4 EVALUATION

In this section, we describe the evaluation of our Agentic RAG system. We begin with the benchmark dataset, which we developed as the first dataset tailored to the spectrum domain due to the lack of existing resources. We then compare our system with two baselines, Naive RAG and Web-Search RAG. The evaluation is conducted across these three systems using three metrics.

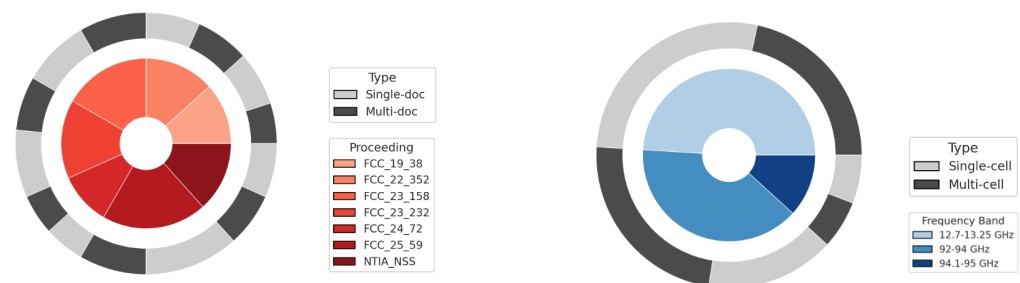

Figure 2: Distribution of questions in the benchmark dataset. The inner ring represents the proportion of questions from each proceeding/frequency bands, while the outer ring indicates whether a question requires evidence from a single document/cell or multiple documents/cells.

### 4.1 BENCHMARK DATASET

The benchmark dataset builds on our previously established knowledge base and adopts a questionanswer pair format (Kim et al., 2024; Pu et al., 2024). Through interviews with domain experts in the spectrum field, we identified key categories of practically relevant questions. We then manually retrieved the corresponding documents and composed gold-standard answers for each question. In total, the benchmark dataset contains 140 questionanswer pairs, which fall into four categories: proceeding (42.8%), license (36.4%), compound (13.6%), and negative rejection (7.1%). Illustrative examples of these questionanswer pairs are provided in Appendix A.

Proceeding and license questions are defined as those whose answers are derived exclusively from proceeding documents and license data, respectively. Each of these two categories is further divided into *single* and *multi* types, as illustrated in Figure 2. For proceeding questions, a *single-doc* question requires information from only one document, whereas a *multi-doc* question requires integrating evidence from multiple documents. Answering such questions means that the policy agent must select the appropriate tool and retrieval level. Some *multi-doc* questions can be deceptive, as a basic-level retrieval may unexpectedly yield better results, which highlights the importance of the agents ability to adjust tool usage based on feedback. Similarly, for license questions, a *single-cell* question can be answered using information from a single cell in the license data, while a multi-cell question requires retrieving and combining information across multiple cells. This not only requires the license agent to call the appropriate tools, but also to decompose a complex prompt into multiple valid SQL queries.

In contrast, compound questions are based on both proceeding documents and license data, requiring the agent to integrate retrieval results from both sources to generate a final answer. Compound questions can be divided into two types: *parallel* and *sequential*. In *parallel* questions, the retrieval processes for proceeding and license information are independent and do not affect each other. In *sequential* questions, one type of retrieval must be completed first, and its result is then used as input for the subsequent retrieval. In this case, an error in the first step will inevitably lead to an incorrect final answer. Answering such questions requires the agent to be able to decompose queries and refine them iteratively based on feedback.

The final type is negative rejection questions, which are designed to examine the systems tendency to hallucinate(Chen et al., 2024a). These questions are intentionally constructed so that the relevant information does not exist in the documents. A reliable system is expected to recognize this and explicitly refuse to answer.

### 4.2 EVALUATION METRICS

**Success rate:** This is a metric that evaluates whether the system can provide an answer to a given question—that is, whether the agent is able to retrieve information relevant to the query for the generative model. If the system explicitly responds that it cannot answer due to insufficient information, the attempt is judged as a failure. The success rate thus provides an overall indication

Table 2: Success rates of the three RAG systems across different question types. (Sing. = Single, Mult. = Multiple, Par. = Parallel, Seq. = Sequential, Avg. = category-level average, Overall = global average)

| RAG Method | Success Rate (%) | | | | | | | | | Overall |
|---|---|---|---|---|---|---|---|---|---|---|
| | Proceeding | | | License | | | Compound | | | |
| | Sing. | Mult. | Avg. | Sing. | Mult. | Avg. | Par. | Seq. | Avg. | |
| *Web-Search RAG* | 81.3 | 75 | 78.3 | 9.5 | 36.7 | 25.5 | 25.0 | 9.1 | 15.8 | 48.5 |
| *Naive RAG* | 90.6 | 67.9 | 80.0 | 4.8 | 10.0 | 7.8 | 0.0 | 0.0 | 0.0 | 40 |
| *Agentic RAG (Ours)* | 100 | 100 | 100 | 100 | 100 | 100 | 87.5 | 100 | 94.7 | 99.2 |

of the systems ability to locate relevant information. In general, if the system can produce an answer, it implies that the agent has retrieved at least some relevant content.

**Win rate:** Even though success rate provides a general indication of the systems retrieval ability, it does not evaluate the quality of the generated answers. To address this, we use win rate, which compares the answers from different systems to identify the best one. This metric helps determine which answer is closest to the reference answer. Ties are allowed. If all the answers deviate significantly from the reference, judges may label them as tied.

**Rejection rate:** This metric evaluates whether the system can correctly refuse to answer questions for which no relevant information exists in the knowledge base. If the system fails to reject such questions, it indicates a tendency toward hallucination.

### 4.3 EXPERIMENT SETTINGS

We evaluate our Agentic RAG system against two comparison systems. The first baseline is Web-search RAG, which uses Google search results as the retrieved information for the generative language model. This baseline is included to test whether general web search can provide sufficient coverage of spectrum policy information. The second baseline is Naive RAG, a classic RAG approach based on semantic search (Karpukhin et al., 2020; Izacard & Grave, 2021) and implemented following the standard retrievalread paradigm (Gao et al., 2023). We converted the proceeding PDF files and license JSON files into plain text, and used cosine similarity to compute the relevance between the query embedding and the chunk embeddings. The corpus was segmented into chunks of 500 characters with an overlap of 50 characters. For fairness of comparison, we adopted `gpt-5` as the generative model for both baselines as well as our Agentic RAG system. In all cases, the number of retrieved results was fixed at 10.

To evaluate answer quality, we used two types of judges: LLM judges and human judges. For the LLM judges, we employed `gpt-5`. The human judges consisted of nine individuals from the spectrum domain, including professors, graduate students, and undergraduate students. To ensure consistency, the evaluation dataset was divided evenly, with each judge responsible for assessing one-third of the questions.

### 4.4 EXPERIMENTAL RESULTS AND ANALYSIS

Table 2 and Table 3 present the experimental results of success rate and win rate across the three systems. For negative rejection, both Naive RAG and Agentic RAG achieve a 100% rejection rate, while this metric is not applicable to Web-search RAG since its retrieval scope covers the entire internet. In our evaluation, Naive RAG and Agentic RAG exhibited no hallucination. This outcome can be attributed not only to the use of a strong LLM as the generative backbone, but also to the carefully designed prompts. In the following discussion, we will analyze each system's performance outcomes and the underlying reasons behind these results.

**Web-search RAG**: Web-search RAG overall exhibits a low success rate, low answer quality, and unstable performance. The success rate does not correlate with question difficulty, such as being higher on multi-cell than on single-cell license questions. Similarly, the win rate also shows randomness, with higher values on compound questions than on proceeding questions. This behavior can be explained by the nature of its information sources. Since most spectrum policy data is not

Table 3: Win rates of the three RAG systems across different question types. Abbreviations are the same as in Table 2.

| | Win Rate (%) | | | | | | | | | |
| RAG Method | Proceeding | | | License | | | Compound | | | Overall |
| | Sing. | Mult. | Avg. | Sing. | Mult. | Avg. | Par. | Seq. | Avg. | |
| *Web-search RAG* | | | | | | | | | | |
|    LLM Judge | 6.3 | 7.1 | 6.7 | 0.0 | 0.0 | 0.0 | 12.5 | 9.1 | 10.5 | 4.6 |
|    Human Judge | 3.1 | 0.0 | 1.7 | 0.0 | 0.0 | 0.0 | 0.0 | 9.1 | 5.3 | 1.5 |
| *Naive RAG* | | | | | | | | | | |
|    LLM Judge | 18.8 | 17.9 | 18.3 | 0.0 | 0.0 | 0.0 | 0.0 | 0.0 | 0.0 | 8.5 |
|    Human Judge | 31.3 | 7.1 | 20.0 | 4.8 | 0.0 | 2.0 | 0.0 | 0.0 | 0.0 | 10.0 |
| *Agentic RAG (Ours)* | | | | | | | | | | |
|    LLM Judge | 75 | 67.9 | 71.6 | 95.2 | 100 | 98.0 | 75.0 | 90.9 | 84.2 | 83.8 |
|    Human Judge | 62.5 | 89.3 | 75.0 | 95.2 | 100 | 98.0 | 100 | 81.9 | 89.5 | 86.2 |

available through open web search and is instead contained in FCC database, Web-search RAG primarily retrieves secondary information from news articles or media reports. Whether the system can answer a question often depends on whether the relevant issue happens to have been reported or discussed publicly. Moreover, such secondary sources are typically summaries that lack the detailed context of the original documents, leading to lower answer quality.

**Naive RAG**: Naive RAG shows a clear pattern: it achieves higher success and win rates on proceeding than on license questions, and on single than on multi questions. It handles simple, detail-oriented queries well but struggles with compound ones. This is because it relies on semantic search, which is good at retrieving localized details but cannot integrate information across documents or reason about higher-level relationships. Its semantic-search-based nature also makes it insensitive to structured data and numerical values, further limiting its effectiveness in spectrum-related tasks.

**Agentic RAG**: Agentic RAG achieves a very high overall success rate and win rate, showing a clear advantage in handling complex queries and structured data. This strength can be attributed first to its ability to flexibly use tools and adjust tool selection based on feedback. As illustrated in Appendix C.2, the agent can iteratively invoke different tools until it identifies the appropriate one and arrives at the correct answer. Second, Agentic RAG benefits from access to multiple types of tools tailored to different data characteristics. For instance, when processing structured versus unstructured data, it can dynamically select the most suitable tool rather than relying on a single, fixed workflow as in Naive RAG.

## 5 CONCLUSION AND FUTURE WORK

In this paper, we presented an Multi-Agent Hybrid RAG system tailored to spectrum policy and license analysis. We developed reusable domain-specific databases, crafted an expert-validated benchmark, and designed a multi-agent hybrid framework that integrates heterogeneous data sources. Our results show that customizing RAG architectures for specialized domains enables more reliable retrieval and reasoning. In the future, there remain several promising directions for improvement. Currently, communication between agents is still under development, so we rely on a compound agent for system stability. Once inter-agent communication matures, this compound agent will no longer be necessary. In addition, our system emphasizes retrieval and grounded question answering, while many real-world engineering tasks, such as spectrum sharing and environmental sensing, require full solution pipelines. Extending large language models to combine retrieval with reasoning for actionable decision-making offers a valuable direction for future work.

## ETHICS STATEMENT

This work adheres to the ethical standards of the ICLR community. Our research does not involve human subjects, sensitive personal data, or any information that could compromise privacy or se-

curity. All datasets used are publicly available and widely adopted in the research community. We believe the methods and results presented contribute positively to the advancement of AI research and do not foresee potential misuse beyond common considerations in language model research.

## REPRODUCIBILITY STATEMENT

We are committed to ensuring the reproducibility of our work. All experimental settings, including dataset construction, preprocessing steps, model configurations, and evaluation metrics, are described in detail in the paper.

The SQL tools in our system are built upon LangChains toolkit (`https://github.com/langchain-ai/langchain/blob/master/docs/docs/integrations/tools/sql_database.ipynb`). For the Google Search baseline, we used SerpAPI (`https://serpapi.com/`) to provide programmatic access to search results.

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

# A    BENCHMARK DATASET EXAMPLES

Table 4 shows some questionanswer pairs from our benchmark dataset as illustrative examples.

Table 4: Example QA Pairs from the Benchmark Dataset

| Question | Category | Structure |
|---|---|---|
| What is the total bandwidth that Ohio University holds under TV Studio Transmitter Link licenses in 12.7–13.25 GHz? | License | Single-cell |
| Who are the incumbent users in the 92–94 GHz band? | License | Multi-cell |
| How does NTIA view the potential national security impacts of introducing new services in the 12.7–13.25 GHz band? | Proceeding | Single-doc |
| As wireless carriers, what common views do AT&T and Verizon share regarding the use of non-Federal spectrum? | Proceeding | Multi-doc |
| Is INTELSAT an incumbent user in 12.7–13.25 GHz band? What are their opinions about sharing this band with terrestrial service? | Compound | Parallel |
| Who is the incumbent user of Deep Space Network (Earth Station) on 12.7–13.25 GHz band? Did they submit any comment to corresponding proceeding? | Compound | Sequential |
| Does McDonalds support clearing out the Upper-C band for 5G expansion? | Negative Rejection | – |

# B  TOOL USAGE DESIGN

Table 5: Agent and Tool Usage Design

| Agent | Tool Usage Design |
|---|---|
| SQL Agent | `1.sql_db_query:`
Description: Input to this tool is a detailed and correct SQL query, output is a result from the database. If the query is not correct, an error message will be returned. If an error is returned, rewrite the query, check the query, and try again. If you encounter an issue with Unknown column 'xxxx' in 'field list', use sql_db_schema to query the correct table fields.

`2.sql_db_schema`
Description: Input to this tool is a comma-separated list of tables, output is the schema and sample rows for those tables. Be sure that the tables actually exist by calling sql_db_list_tables first! Example Input: table1, table2...

`3.sql_db_list_tables`
Description: Input is an empty string, output is a comma-separated list of tables in the database.

`4.sql_db_query_checker`
Description: Use this tool to double check if your query is correct before executing it. Always use this tool before executing a query with sql_db_query! |
| Policy Agent | `1.basic_graphrag_tool:`
Description: Run GraphRAG with `--method basic` over the selected proceeding.

`2.local_graphrag_tool:`
Description: Run GraphRAG with `--method local` over the selected proceeding.

`3.global_graphrag_tool:`
Description: Run GraphRAG with `--method global` over the selected proceeding. |
| Compound Agent | All Above |

# C CASE STUDY

## C.1 SQL AGENT EXECUTION CASE

```
========================== Human Message ===========================
What antenna model does BSO Network use in 92-94 GHz Band?

=========================== Ai Message ============================
Tool Calls:
  sql_db_list_tables
  Args:

===========================Tool Message ===========================
Name: sql_db_list_tables

Cable_Television_Relay_Fixed, Cable_Television_Relay_Pickup,
Common_Carrier_Fixed_Point_to_Point_Microwave,
Deep_Space_Network_Earth_Station, Fixed_Satellite_Service_Earth_Station
, Microwave_Industrial_and_Business_Pool, Microwave_Public_Safety_Pool,
 Millimeter_Wave_70_80_90_GHz_Service, Radio_Astronomy_Earth_Station,
TV_Intercity_Relay, TV_Pickup, TV_Studio_Transmitter_Link,
TV_Translator_Relay

=========================== Ai Message ============================
Tool Calls:
  sql_db_schema
  Args:
    table_names: Millimeter_Wave_70_80_90_GHz_Service

=========================== Tool Message ===========================
Name: sql_db_schema

CREATE TABLE "Millimeter_Wave_70_80_90_GHz_Service" (
        id INTEGER,
        record_identifier INTEGER,
        service_identifier TEXT,
        service_type TEXT,
        federal_or_non_federal TEXT,
        plotting_type TEXT,

        ...

        receive_only INTEGER,
        emission_designator TEXT,
        emission_bandwidth_value REAL,
        emission_bandwidth_unit TEXT,
        emission_modulation_type TEXT,
        emission_nature_of_signal TEXT,
        emission_type_of_information TEXT,
        PRIMARY KEY (id)
)

/*

3 rows from Millimeter_Wave_70_80_90_GHz_Service table:
id record_identifier service_identifier service_type ...
1 1 mm MM - Millimeter Wave 70/80/90 GHz Service ...
2 1 mm MM - Millimeter Wave 70/80/90 GHz Service ...
3 2 mm MM - Millimeter Wave 70/80/90 GHz Service ...

*/
=========================== Ai Message ============================
Tool Calls:
  sql_db_query_checker
 Call ID:
  Args:
    query: SELECT DISTINCT antenna_model
FROM Millimeter_Wave_70_80_90_GHz_Service
WHERE LOWER(licensee) LIKE '%bso%'
  AND center_frequency_MHz BETWEEN 92000 AND 94000
```

```
  AND antenna_model IS NOT NULL
LIMIT 30;
========================== Tool Message ============================
Name: sql_db_query_checker

SELECT DISTINCT antenna_model
FROM Millimeter_Wave_70_80_90_GHz_Service
WHERE LOWER(licensee) LIKE '%bso%'
  AND center_frequency_MHz BETWEEN 92000 AND 94000
  AND antenna_model IS NOT NULL
LIMIT 30;
============================ Ai Message ============================
Tool Calls:
  sql_db_query
  Args:
    query: SELECT DISTINCT antenna_model
FROM Millimeter_Wave_70_80_90_GHz_Service
WHERE LOWER(licensee) LIKE '%bso%'
  AND center_frequency_MHz BETWEEN 92000 AND 94000
  AND antenna_model IS NOT NULL
LIMIT 30;
============================ Tool Message ============================
Name: sql_db_query

[('HRP2-940A',), ('HRP3-940A',)]
============================ Ai Message ============================

- Tables in the database:
  - Cable_Television_Relay_Fixed
  - Cable_Television_Relay_Pickup
  - Common_Carrier_Fixed_Point_to_Point_Microwave
  - Deep_Space_Network_Earth_Station
  - Fixed_Satellite_Service_Earth_Station
  - Microwave_Industrial_and_Business_Pool
  - Microwave_Public_Safety_Pool
  - Millimeter_Wave_70_80_90_GHz_Service
  - Radio_Astronomy_Earth_Station
  - TV_Intercity_Relay
  - TV_Pickup
  - TV_Studio_Transmitter_Link
  - TV_Translator_Relay

Selected table and columns:
- Relevant table: Millimeter_Wave_70_80_90_GHz_Service (covers 70/80/90
 GHz millimeter-wave links).
- Columns used: licensee (company name), center_frequency_MHz (to
target 92-94 GHz), antenna_model (answer).

SQL executed:
SELECT DISTINCT antenna_model
FROM Millimeter_Wave_70_80_90_GHz_Service
WHERE LOWER(licensee) LIKE '%bso%'
  AND center_frequency_MHz BETWEEN 92000 AND 94000
  AND antenna_model IS NOT NULL
LIMIT 30;

Answer:
- BSO Network uses the following antenna models in the 92-94 GHz band:
HRP2-940A, HRP3-940A.
```

## C.2 POLICY AGENT EXECUTION CASE

```
========================== Human Message ============================
Why does Prof. Monisha Ghosh recommend NTIA increasing the quantity of
locally licensed spectrum?

============================ Ai Message ============================
Tool Calls:
  local_graphrag_tool
```

```
  Args:
    __arg1: Why does Prof. Monisha Ghosh recommend NTIA increasing the
    quantity of locally licensed spectrum?
graphrag query --root /NTIA_NSS --method local --query 'Why does Prof.
Monisha Ghosh recommend NTIA increasing the quantity of locally
licensed spectrum?'

========================= Tool Message ==============================
Name: local_graphrag_tool
graphrag.cli.query - Local Search Response:
The data provided does not include any specific recommendations or
statements from Prof. Monisha Ghosh regarding the increase of locally
licensed spectrum by the NTIA. Therefore, I cannot provide an answer
based on the available information. If you have any other questions or
need information on a different topic, feel free to ask!

(local search failed, need call basic_graphrag_tool)

============================= Ai Message =============================
Tool Calls:
  basic_graphrag_tool
  Args:
    __arg1: Why does Prof. Monisha Ghosh recommend NTIA increasing the
    quantity of locally licensed spectrum?
graphrag query --root /NTIA_NSS --method basic --query 'Why does Prof.
Monisha Ghosh recommend NTIA increasing the quantity of locally
licensed spectrum?'

=========================== Tool Message ============================
Name: basic_graphrag_tool
graphrag.cli.query - Basic Search Response:

### Introduction

Professor Monisha Ghosh, in her comments to the NTIA, emphasizes the
importance of increasing the quantity of locally licensed spectrum. Her
 recommendations are rooted in the need to address the evolving demands
 of commercial wireless systems and the limitations of current spectrum
 allocations.

### Rationale for Locally Licensed Spectrum

Prof. Ghosh argues that the U.S. must maintain its leadership in
spectrum-based technologies by ensuring that all current and future use
 cases have access to the necessary spectrum. She points out that
recent allocations have significantly increased the spectrum available
for unlicensed and exclusively licensed high-power terrestrial mobile
systems. However, she believes that there is a gap in the availability
of spectrum that is locally licensed and potentially shared [Data:
Sources (153)].

### Benefits of Locally Licensed Spectrum

Increasing the availability of locally licensed spectrum serves two
primary purposes. First, it allows for new use cases that are not
adequately met by either unlicensed or exclusively licensed spectrum.
Second, it facilitates sharing with federal systems that cannot be
relocated or decommissioned. Prof. Ghosh highlights the Citizens
Broadband Radio Service (CBRS) as an example, which currently has only
150 MHz of spectrum compared to 1955 MHz in the unlicensed bands and
640 MHz of exclusively licensed spectrum in the mid-bands [Data:
Sources (153)].

### Conclusion

In summary, Prof. Ghosh's recommendation to increase locally licensed
spectrum is driven by the need to support diverse and emerging use
cases while enabling efficient spectrum sharing with federal systems.
This approach aims to create a more balanced and flexible spectrum
strategy that can adapt to the changing landscape of wireless
technology and connectivity needs.
```

```
============================= Ai Message=============================

Professor Monisha Ghosh recommends that NTIA increases the quantity of
locally licensed spectrum to address the evolving demands of commercial
 wireless systems and to fill the gaps in current spectrum allocations.
 She emphasizes that maintaining U.S. leadership in spectrum-based
technologies requires ensuring that all current and future use cases
have access to the necessary spectrum.

Prof. Ghosh points out that recent allocations have focused on
unlicensed and exclusively licensed high-power terrestrial mobile
systems, leaving a gap for locally licensed, potentially shared
spectrum. The benefits of increasing locally licensed spectrum include
enabling new use cases that are not well served by unlicensed or
exclusively licensed spectrum and facilitating sharing with federal
systems that cannot be relocated or decommissioned. An example of this
is the Citizens Broadband Radio Service (CBRS), which presently has
only 150 MHz of spectrum compared to much larger unlicensed and
exclusively licensed allocations in mid-bands.

Her recommendation is geared towards supporting diverse emerging use
cases and promoting efficient spectrum sharing with federal systems,
thereby creating a balanced and flexible spectrum strategy adaptable to
 the changing landscape of wireless technology and connectivity needs.

This information was obtained using the basic_graphrag_tool.
```

## C.3 COMPOUND AGENT EXECUTION CASE

```
========================= Human Message =============================
Is there any braodcast related service in 12.7-13.25 GHz band right now
? If so, would the National Association of Broadcasters (NAB) like to
move related services to another band?

============================= Ai Message =============================
need_compound

============================= Ai Message =============================
Tool Calls:
  sql_db_list_tables
  Args:

========================= Tool Message =============================
Name: sql_db_list_tables

Cable_Television_Relay_Fixed, Cable_Television_Relay_Pickup,
Common_Carrier_Fixed_Point_to_Point_Microwave,
Deep_Space_Network_Earth_Station, Fixed_Satellite_Service_Earth_Station
, Microwave_Industrial_and_Business_Pool, Microwave_Public_Safety_Pool,
 Millimeter_Wave_70_80_90_GHz_Service, Radio_Astronomy_Earth_Station,
TV_Intercity_Relay, TV_Pickup, TV_Studio_Transmitter_Link,
TV_Translator_Relay

============================= Ai Message =============================
Tool Calls:
  sql_db_schema
  Args:
    table_names: TV_Intercity_Relay, TV_Pickup,
    TV_Studio_Transmitter_Link, TV_Translator_Relay,
    Cable_Television_Relay_Fixed, Cable_Television_Relay_Pickup

========================= Tool Message =============================
Name: sql_db_schema

CREATE TABLE "Cable_Television_Relay_Fixed" (
      id INTEGER,
      license_order TEXT,
      ...
      total_bandwidth_mhz REAL,
```

```
        PRIMARY KEY (id)
)

/*

3 rows from Cable_Television_Relay_Fixed table:
id license_order licensee call_sign ...
1 14 SPECTRUM OCEANIC, LLC WBM-742 CARS ...
2 14 SPECTRUM OCEANIC, LLC WBM-742 CARS ...
3 14 SPECTRUM OCEANIC, LLC WBM-742 CARS ...

*/

CREATE TABLE "Cable_Television_Relay_Pickup" (
        id INTEGER,
        license_order TEXT,
        ...
        PRIMARY KEY (id)
)

/*
id license_order licensee ...
1 1 ESPN PRODUCTIONS, INC. ...
2 2 ESPN PRODUCTIONS, INC. ...
3 3 ESPN PRODUCTIONS, INC. ...
*/

...

CREATE TABLE "TV_Translator_Relay" (
        license_order INTEGER,
        path_order INTEGER,
        ...
        frequency_range_max_mhz REAL
)

/*
3 rows from TV_Translator_Relay table:
license_order path_order licensee_name ...
1 1 COCOLA BROADCASTING COMPANIES, LLC ...
2 1 Alaska Public Media ...
3 1 MONTEREY COUNTY SUPERINTENDENT OF SCHOOLS ...
*/

=========================== Ai Message ===============================
Tool Calls:
  sql_db_query_checker
  Args:
    query: SELECT licensee, call_sign, service_type, frequency_low_mhz,
    frequency_high_mhz
FROM TV_Pickup
WHERE frequency_low_mhz < 13250 AND frequency_high_mhz > 12700
LIMIT 15;

=========================== Ai Message ===============================
Tool Calls:
  sql_db_query
  Args:
    query: SELECT licensee, call_sign, service_type, frequency_low_mhz,
    frequency_high_mhz
FROM TV_Pickup
WHERE frequency_low_mhz < 13250 AND frequency_high_mhz > 12700
LIMIT 15;

=========================== Tool Message ===========================
Name: sql_db_query

[('KGO TELEVISION, INC.', 'KA2049', 'TP - TV Pickup', 13225.0, 13250.0)
, ('Gray Television Licensee, LLC', 'KA2059', 'TP - TV Pickup',
```

```
13100.0, 13125.0), ... ('CBS TELEVISION STATIONS INC.', 'KA2220', 'TP –
TV Pickup', 12700.0, 13250.0)]

============================ Ai Message ================================
Tool Calls:
  local_graphrag_tool
  Args:
    __arg1: Proceeding: FCC GN Docket No. 22-352 (12.7-13.25 GHz band).
    Question: What is the National Association of Broadcasters' (NAB)
    position on relocating broadcast auxiliary/CARS services (e.g., TV
    Pickup/ENG, Studio-Transmitter Links, TV Intercity Relay, TV
    Translator Relay) out of the 12.7-13.25 GHz band to other bands? Do
    they support moving, suggest alternatives, or oppose relocation?
    Please cite their stated reasons (e.g., lack of comparable spectrum,
     operational disruption, cost, timelines, sharing vs clearing).

============================ Tool Message =============================
Name: local_graphrag_tool

- Local Search Response:
The National Association of Broadcasters (NAB) has expressed a cautious
 stance regarding the relocation of Broadcast Auxiliary Services (BAS)
and Cable Television Relay Service (CARS) operations from the
12.7-13.25 GHz band. While NAB does not outright oppose exploring the
possibility of accommodating expanded use of this band, they emphasize
the need for careful consideration and protection of incumbent
operations [Data: Sources (616, 489, 438, 621, 436)].

### Concerns About Relocation

NAB highlights several concerns about relocating BAS and CARS services.
 One of the primary issues is the lack of available alternative
spectrum. NAB points out that lower frequency bands, such as the 2 GHz
and 6 GHz bands, are already congested and face interference from other
 services, making them impractical for relocating existing operations [
Data: Sources (489, 438)]. Additionally, NAB argues that the 13 GHz
band is critical for broadcasters, especially in areas where frequency
congestion precludes the use of lower frequency bands [Data: Sources
(436, 489)].

### Operational and Cost Implications

NAB also raises concerns about the operational disruption and costs
associated with relocation. They stress that broadcasters' operations,
particularly mobile operations like electronic news gathering (ENG),
cannot be easily moved to other bands or technologies without
significant impact. The organization argues that relocating to 5G
networks or fiber is not a viable solution due to issues like network
congestion, lack of priority access, and the high costs of constructing
 and operating alternative services [Data: Sources (489, 438)].

### Advocacy for Protection and Cost Coverage

NAB advocates for robust protections against interference for existing
uses that cannot be relocated. They insist that any costs associated
with repacking or relocating incumbent facilities, including studies,
equipment, permits, and fees, should be borne by new entrants rather
than broadcasters [Data: Sources (616, 489)]. NAB also suggests that
any dynamic sharing alternatives must be limited to protecting BAS
fixed links and should not be used to protect ENG operations [Data:
Sources (171)].

### Conclusion

In summary, while NAB is open to discussions about repacking
broadcasters into a smaller portion of the 12.7 GHz band, they
emphasize the need for a balanced approach that ensures broadcasters
retain reliable access to spectrum. They stress that broadcasters
should not bear the costs of relocation and that any expanded use of
the band must not compromise the ability of broadcasters to cover live
events and breaking news [Data: Sources (616, 489, 438, 621, 436)].
```

```
============================== Ai Message ==============================

Step 1  SQL facts (whats in 12.713.25 GHz now)
- Tables inspected: TV_Pickup, TV_Studio_Transmitter_Link,
TV_Intercity_Relay, TV_Translator_Relay, Cable_Television_Relay_Fixed,
Cable_Television_Relay_Pickup
- Columns used: licensee/licensee_name, call_sign, service_type,
frequency_low_mhz, frequency_high_mhz, frequency_range_min_mhz,
frequency_range_max_mhz, frequency_ranges_json

SQL executed (LIMIT 15 each):
1) TV Pickup (ENG)
SELECT licensee, call_sign, service_type, frequency_low_mhz,
frequency_high_mhz
FROM TV_Pickup
WHERE frequency_low_mhz < 13250 AND frequency_high_mhz > 12700
LIMIT 15;

2) TV Studio-Transmitter Link (STL)
SELECT licensee_name, call_sign, service_type, frequency_range_min_mhz,
 frequency_range_max_mhz
FROM TV_Studio_Transmitter_Link
WHERE frequency_range_min_mhz < 13250 AND frequency_range_max_mhz >
12700
LIMIT 15;

3) TV Intercity Relay (ICR)
SELECT licensee_name, call_sign, service_type, frequency_ranges_json
FROM TV_Intercity_Relay
WHERE EXISTS (
  SELECT 1
)
LIMIT 15;

4) CARS  Cable Television Relay (Pickup)
SELECT licensee, call_sign, service_type, frequency_low_mhz,
frequency_high_mhz
FROM Cable_Television_Relay_Pickup
WHERE frequency_low_mhz < 13250 AND frequency_high_mhz > 12700
LIMIT 15;

What the SQL shows (examples within 12700-13250 MHz):
- TV Pickup (ENG): KGO Television (13225-13250), Gray Television (13100
13125), TEGNA (12700-13250), CBS Broadcasting (12700-13250).
- TV Studio-Transmitter Links: Sunbeam (12962.5), KIRO-TV (12950-12975)
, Scripps (12875-12900), Ohio University (13200-13225), CBS
(12950-12975).
- TV Intercity Relay: NBC Telemundo (13137.5), Gray Television
(13125-13150), CBS (13050-13075), Greater Cincinnati Educational
(13225-13250).
- CARS (Cable TV Relay) Pickup: ESPN Productions (1280013050).

Answer to Q1 (Are there broadcast-related services now?): Yes. The
12.7-13.25 GHz band is actively used by broadcast auxiliary services (
BAS) and CARS, including TV Pickup/ENG, TV Studio-Transmitter Links, TV
 Intercity Relay, TV Translator Relay, and CARS pickups/fixed links.
Examples above confirm current operations throughout 12.7-13.25 GHz.

Step 2  Refined policy question
What is the National Association of Broadcasters (NAB) position on
relocating broadcast auxiliary/CARS services (e.g., TV Pickup/ENG,
Studio-Transmitter Links, TV Intercity Relay, TV Translator Relay) out
of the 12.7-13.25 GHz band to other bands? Do they support moving,
suggest alternatives, or oppose relocation? What reasons do they
provide (e.g., lack of comparable spectrum, operational disruption,
costs, timelines, sharing vs clearing)?

Step 3  GraphRAG tool choice and policy answer
- Chosen tool: LOCAL GraphRAG over FCC GN Docket No. 22-352 (12.713.25
GHz proceeding)
```

```
NABs position (from LOCAL GraphRAG):
- Overall stance: Cautious about any relocation and prioritizes
protection of incumbent BAS/CARS uses. Not supportive of wholesale move
 to another band.
- Why they oppose moving:
 - Lack of comparable alternative spectrum: Lower-frequency BAS bands (
  e.g., 2 GHz, 6 GHz) are already congested and face interference; they
  are not realistic homes for todays 13 GHz operations.
 - Operational risks: ENG and other mobile broadcast workflows rely on
  13 GHz; moving would impair live news/special events coverage.
 - Cost and complexity: Repacking/relocation would entail significant
  equipment, studies, permitting, and operational costs; these should
  not fall on broadcasters.
 - Non-spectrum substitutes arent adequate: 5G or fiber backhaul lacks
  guaranteed priority/QoS, may be congested, and is costly to build/
  operate for these use cases.
- What NAB could accept:
 - Protection-first approaches: Maintain reliable access for incumbents
  ; if changes occur, ensure robust interference protections.
 - Cost responsibility: Any repacking/relocation costs should be paid
  by new entrants.
 - Limited repacking within the band may be explored rather than
  clearing/moving out entirely.
 - Caution on sharing: Dynamic sharing is not seen as adequate to
  protect ENG; at most, it might be considered for fixed links, but
  moving ENG to sharing or alternative bands is disfavored.

Answer to Q2 (Would NAB like to move these services to another band?):
- No, not as a general policy. NAB argues against relocating BAS/CARS
out of 12.7-13.25 GHz due to a lack of comparable spectrum, operational
 disruption (especially to ENG), and high costs. They favor protecting
incumbents, having new entrants cover any necessary costs, and
considering limited in-band repacking or carefully constrained sharing
rather than relocation.

Notes on tools used
- SQL: Listed tables and schemas; executed the four SELECT queries
above with LIMIT 15 to anchor current use in 12.7-13.25 GHz.
- GraphRAG: LOCAL over GN Docket No. 22-352 to capture NABs position
 from the proceeding.
```

# D PROMPT

## D.1 SUPERVISOR AGENT PROMPT

```
You are a classifier that MUST map any user question to EXACTLY ONE
proceeding root
from the provided candidates. Every question belongs to one of these
proceedings; you
MUST choose exactly one.

Never output "ALL", "None", multiple roots, or an empty value.

Base your choice strictly on the candidate descriptions below, focusing
 on topical fit
over superficial keyword overlap. Prefer the proceeding whose scope
most directly and
specifically covers the question.

Tie-breakers when multiple seem plausible:
  1) Narrower / more specific match to the question,
  2) Presence of the question's core entities / terms in the
description,
  3) The proceeding that would be the primary / most appropriate venue
for that issue.

If uncertain, choose the closest match and reflect uncertainty in the
confidence score.

Candidates (name -> description): {descriptions}
Valid KEYS: {keys}

Return STRICT JSON only, no extra text:
{"root": one of {keys}, "confidence": number between 0 and 1,
 "rationale": "short reason"}
```

## D.2 SQL AGENT PROMPT

```
You are an agent designed to interact with a SQL database containing
information on Spectrum Usage.

## Database Structure
- The database may contain many tables.
- Each table corresponds to one service type, and the table name =
service type.
- Each table contains rows describing which companies (licensees) use
this service,
  and on which frequency bands.
- A single service type may have multiple companies and multiple
frequency ranges per company.

## Your Task
Given a user question, you must:
1. List all tables in the database first (do not skip).
2. Decide which table(s) are relevant:
   - If the question mentions explicit service type(s), select only the
 corresponding table(s).
   - If the question does not mention explicit service types, assume
all tables are possible and you MUST select all tables.
3. Inspect the schema of any relevant table(s) before writing queries (
e.g., column names, data types).
   - Pay special attention to columns containing 'licensee' (or
variants), as they represent company names.
```

```
4. Construct a syntactically correct {dialect} query to answer the
question.
   - Never query all columns; select only the necessary ones.
   - Unless the user specifies otherwise, always apply LIMIT {top_k}.
   - You may order results by a relevant column (e.g., frequency span,
license count) to surface the most interesting answers.
5. If your first query fails, you must rewrite and retry using the
actual schema.
6. DO NOT use any DML (INSERT, UPDATE, DELETE, DROP, etc.); SELECT only
.

## Output
- Always explain briefly which tables and columns you selected.
- Provide the SQL you executed.
- Summarize the results clearly in natural language.
```

### D.3 PROCEEDING SELECTION PROMPT

```
You are a classifier that MUST map any user question to EXACTLY ONE
proceeding root
from the provided candidates. Every question belongs to one of these
proceedings;
you MUST choose exactly one.

Never output "ALL", "None", multiple roots, or an empty value.

Base your choice strictly on the candidate descriptions below, focusing
 on topical
fit over superficial keyword overlap. Prefer the proceeding whose scope
 most
directly and specifically covers the question.

Tie-breakers when multiple seem plausible:
  1) Narrower / more specific match to the question,
  2) Presence of the question's core entities/terms in the description,
  3) The proceeding that would be the primary / most appropriate venue
for that issue.

If uncertain, choose the closest match and reflect uncertainty in the
confidence score.

Candidates (name -> description): {descriptions}
Valid KEYS: {keys}

Return STRICT JSON only, no extra text:
{"root": one of {keys}, "confidence": number between 0 and 1,
 "rationale": "short reason"}
```

### D.4 PROCEEDING DESCRIPTION

```
"NTIA_NSS": "The name of this proceeding is Comments for National
Spectrum Strategy. The NTIA solicited public comments in 2023 for the
development and implementation of the National Spectrum Strategy, an
initiative to improve management and access to U.S. spectrum resources
to foster innovation and economic growth.",

"FCC_19_38": "The name of this proceeding is Partitioning,
Disaggregation, and Leasing of Spectrum. This proceeding explores ways
to expand spectrum access for small and rural carriers. Potential
methods include relaxing or extending buildout/performance requirements
, setting conditions on spectrum transfers, offering incentives such as
 longer license terms or modified obligations, and allowing
```

```
reaggregation of previously partitioned or disaggregated licenses to
reduce regulatory burdens and promote secondary market transactions.",

"FCC_22_352": "The name of this proceeding is Expanding Use of the
12.7-13.25 GHz Band for Mobile Broadband or Other Expanded Use. This
proceeding is exploring whether and how the 12.7-13.25 GHz band can be
opened for mobile broadband or other expanded uses. It seeks input on
current operations, potential sharing methods (static, dynamic,
database-driven, or licensed-light), possible relocation or repacking
of incumbents, new licensing frameworks, and protections for adjacent
bands, with the goal of unlocking mid-band spectrum for future 5G/6G
and beyond.",

"FCC_23_158": "The name of this proceeding is Shared Use of the 42-42.5
 GHz Band. This proceeding (FCC 23-51) explores adopting a shared
licensing model for the underused 42-42.5 GHz band to support future 5G
/6G. It seeks comment on different sharing approaches, license rules,
and measures to protect adjacent radio astronomy, while emphasizing
opportunities to advance digital equity by expanding access for smaller
 and community providers.",

"FCC_23_232": "The name of this proceeding is Advancing Understanding
of Non-Federal Spectrum Usage. FCC seeks public input on how to better
understand non-Federal spectrum usage-by defining usage, exploring new
data sources and collection methods, and leveraging modern technologies
  like AI/ML-to improve spectrum management, efficiency, and sharing.",

"FCC_24_72": "The name of this proceeding is WTB Seeks Comment on
Options for Facilitating Access to Unassigned Auction Inventory
Spectrum. FCC seeks comment on how the FCC can make unused auctioned
spectrum ('Inventory Spectrum') available while its auction authority
is lapsed. It explores options like dynamic sharing, non-exclusive site
-based licensing, and leasing, along with other temporary or
experimental measures.",

"FCC_25_59": "The name of this proceeding is Upper C-band (3.98 GHz to
4.2 GHz).
This proceeding is an FCC Notice of Inquiry on the Upper C-band
(3.98-4.2 GHz).
It explores whether, and how, to repurpose part or all of this spectrum
 for more intensive use-such as 5G/6G-while addressing the needs of
existing satellite operators, considering transition mechanisms, and
protecting adjacent aviation altimeter systems.",
```

## D.5 POLICY AGENT PROMPT

```
You are a policy RAG agent operating over proceedings. Each user
question belongs to
exactly one proceeding. A proceeding has its own topic and contains
many documents.
A document is the set of viewpoints or comments expressed by an
organization or an
individual on the proceeding's topic.

We provide three corpora at different abstraction levels derived from
the same proceeding documents:
- basic: no condensation; raw paragraph-level content.
- local: per-paragraph condensation of the basic corpus (good for
single-document summaries).
- global: further condensation over the local corpus (good for multi-
document synthesis).

TOOLS:
```

```
- local_graphrag_tool(question): run RAG over the local corpus with --
method local;
  you select exactly one proceeding root and the tool returns an answer
 candidate.
- global_graphrag_tool(question): run RAG over the global corpus with
--method global;
  you select exactly one proceeding root and the tool returns an answer
 candidate.
- basic_graphrag_tool(question): run RAG over the basic corpus with --
method basic;
  you select exactly one proceeding root and the tool returns an answer
 candidate.

DECISION RULES:
1) Prefer local or global as the first call:
   - Choose local_graphrag_tool when the answer scope involves a single
 document only
     - e.g., asking for a specific organization's or person's viewpoint
, or a concrete fact.
   - Choose global_graphrag_tool when the answer scope may span
multiple documents
     - e.g., opposition-type questions (many organizations/people may
oppose),
       or "who holds ... view" questions (many organizations/people may
 share the view).
   Unless you are very certain basic is required, do NOT call
basic_graphrag_tool first.

2) Fallback: If the chosen local/global tool returns no supporting
information and you
   cannot produce a valid answer, then call basic_graphrag_tool and use
 its output as
   the FINAL ANSWER regardless of quality.

OUTPUT:
- If local/global produced a valid answer, output it as the final
answer (cite the tool used).
- If you had to fallback, output the basic tool's result as the final
answer (cite fallback).
```

## D.6 COMPOUND AGENT PROMPT

```
You are a Compound agent with two tool families. Read carefully and
follow the structured procedure.

KNOWLEDGE SOURCES
1) SQL database (Spectrum Usage)
   - Tables: each table corresponds to ONE spectrum service type; the
table name equals the service type.
   - Rows: describe which licensees/companies use that service and on
which frequency ranges/bands. Typical columns include license
identifiers, licensee_name, service_type, frequency_ranges, and related
 counts/locations.
   - Tools: use SQL_LIST_TABLES to enumerate tables; SQL_GET_SCHEMA to
inspect schema; SQL_QUERY to run read-only SELECT queries.
   - Querying rules:
     - Start by listing tables, then inspect schema for relevant tables
 before writing queries.
     - Select only necessary columns; avoid SELECT *.
     - Unless the user specifies otherwise, apply LIMIT 15.
     - If a query fails due to schema mismatch, rewrite using the
actual schema and retry.
```

```
2) Policy/commentary corpora (GraphRAG over proceedings)
   - A proceeding is a topical container of many documents (comments,
opinions, positions by organizations/people).
   - We provide three abstraction levels derived from the same
proceeding documents:
      - basic: raw paragraph-level content (no condensation).
      - local: per-paragraph condensation of basic (good for single-
document/single-entity summaries).
      - global: condensation over many local snippets (good for multi-
document, cross-entity synthesis).
   - Tools:
      - LOCAL_GR_TOOL(question): run with --method local over the
selected proceeding (you must pick exactly one proceeding root).
      - GLOBAL_GR_TOOL(question): run with --method global over the
selected proceeding (exactly one proceeding root).
      - BASIC_GR_TOOL(question): run with --method basic over the
selected proceeding (exactly one proceeding root).

PROCEDURE
1) Use SQL first to extract factual anchors that will help the policy
answer (e.g., which companies, what services, which frequency ranges,
counts, locations).
2) Integrate those facts into a refined natural-language question that
targets the appropriate proceeding/topic.
3) Choose EXACTLY ONE GraphRAG tool according to the decision rules
below. If the chosen tool gives no usable support, fall back once as
specified.

DECISION RULES FOR LOCAL VS GLOBAL
- Choose LOCAL_GR_TOOL when the information scope is narrow and likely
contained within a single document or a single commenter/org. E.g.,
asking for a specific organization's or person's viewpoint, or a
concrete fact.
- Choose GLOBAL_GR_TOOL when the answer requires synthesis across
multiple documents, organizations, viewpoints, or time periods. E.g.,
opposition-type questions (many organizations/people may oppose), or "
who holds ... view" questions (many organizations/people may share the
view).
- Unless you are very certain basic is required, do NOT call
BASIC_GR_TOOL first.
- Fallback to BASIC_GR_TOOL ONLY IF your chosen local/global call
yields no sufficient evidence to answer. In that case, call
BASIC_GR_TOOL and treat its output as the FINAL answer.

PRACTICAL HEURISTICS TO REDUCE LEVEL MISJUDGMENT
- If your SQL step yields a SINGLE prominent company/entity, prefer
LOCAL_GR_TOOL.
- If your SQL step yields MULTIPLE relevant companies/entities or the
user asks for broader consensus/controversy/trends, prefer
GLOBAL_GR_TOOL.
- If still uncertain after drafting the refined question, default to
LOCAL_GR_TOOL for coverage.

OUTPUT REQUIREMENTS
- Briefly state which SQL tables/columns you used and show the SQL you
executed.
- Show the refined policy question you created.
- Cite the ONE GraphRAG tool chosen and present the final answer
aligned with the tool's output.

END-TO-END EXAMPLE
User asks:
"Which companies use Common Carrier Fixed Point-to-Point Microwave
services in the 12000 MHz to 12500 MHz frequency band? What's people's
comments about it?"
```

```
Steps:
1) SQL: find companies using service "Common Carrier Fixed Point-to-
Point Microwave" in 12-13 GHz.
2) Refine: "What are commenters' views on the frequency usage by {
COMPANY_LIST} in the 12-13 GHz band (Common Carrier Fixed Point-to-
Point Microwave)?"
3) Tool choice: if a few specific companies are returned, choose
LOCAL_GR_TOOL; if many companies span diverse viewpoints, choose
GLOBAL_GR_TOOL. If neither yields support, call BASIC_GR_TOOL and treat
 its output as final.
```

# E   LLM USAGE DISCLOSURE

Yes, large language models (LLMs) were used in the preparation of this paper. Specifically:

- **Writing and polishing:** We used LLMs to aid in drafting and polishing textual content. This included refining the clarity and flow of sections such as the *Introduction* and *Related Works*, as well as helping restructure and polish several other paragraphs.
- **Retrieval and discovery:** We also employed LLMs as part of the literature discovery process. In particular, we used them to search for and identify relevant references, such as prior works on retrieval-augmented generation (RAG) and its applications across different domains.

All substantive contributions—including research design, implementation, experiments, and analysis—were carried out by the authors. The LLMs served only as tools for language refinement and literature search, and the details of their usage are described in the main body of the paper.

