# OpenReview forum: "Spectrum Intelligence via Multi-Agent Hybrid Retrieval-Augmented Generation"
_ICLR.cc/2026/Conference — ICLR 2026 Conference Withdrawn Submission_

### Official Review · Reviewer_NuAH · 2025-10-25

**Soundness:** 2
**Presentation:** 3
**Contribution:** 1
**Rating:** 2
**Confidence:** 4

**Summary:**

This paper introduced a multi-agent retrieval-augment generation (RAG) method for question answering in spectrum policy making. Spectrum policy knowledge spans documents that involve precise numerical data and implicit graphical structures. To deal with that, the author crafted an SQL database and graph docs to store the key data, which improves the retrieval efficiency in this task. They also crafted a spectrum policy, making a QA dataset. The method shows improvements compared to the naive websearch method and the basic RAG baselines.

**Strengths:**

1. The paper is easy to follow.
2. The method shows good improvement compared to the naive RAG and web-search agent in the dataset.
3. The authors contribute a new benchmark in this domain.

**Weaknesses:**

1. There is no ablation study on the components, i.e., the different agents. It is hard to know the contribution of each component.
2. The proposed method is tailored for the specific problem of spectrum policy making, which is narrow and not generalizable. The evaluation is also restricted to the small-sized benchmark crafted by the authors.
3. The proposed method involves complicated and task-specific engineering that organizes the data into different specialized formats, such as SQL and Graph, which can be expensive to use and scale up.
4. Overall, the proposed method is only specialized for databases for the given narrow task, and shows little insight into how to make better and generalizable RAG agents. The target application is also relatively narrow, which may better fit its specific community.

**Questions:**

Questions:

1. Why not just wrap these different query methods as different function calls for one agent? Current LLMs (e.g., GPT -5 in the paper) can easily handle such a small number of functions. The design of the router agent seems to be redundant in the context of function calling capacity and the hybrid/reasoning model (GPT-5) today, which can adaptively choose the right tool to use and iteratively call them in their reasoning process.
2. There is little information provided about the human judge.
3. For the RAG baseline, does increasing the chunk size improve performance? It is only 500 tokens for a chunk, which is way smaller than the context length of the latest LLMs. Does increasing the chunk size reduce the performance differences?

Typos:

1. Table 2 -> Naive RAG -> Overall: 40 should be 40.0, to make precision consistent

---

### Official Review · Reviewer_on4L · 2025-10-30

**Soundness:** 2
**Presentation:** 3
**Contribution:** 2
**Rating:** 4
**Confidence:** 3

**Summary:**

This paper proposes a novel hybrid RAG architecture for the complex domain of spectrum policies, effectively integrating structured SQL license data with unstructured policy documents. The primary contribution of this research is to introduce a novel approach for handling heterogeneous data and to construct an expert-validated benchmark dataset.

**Strengths:**

1. The paper introduces a real-world application (spectrum policy) to the RAG community, tackling the complex intersection of legal and technical documents.

2. The proposed hybrid RAG architecture (Sec. 3) for jointly querying heterogeneous data—structured SQL databases and unstructured knowledge graphs.

3. The authors provide a new, expert-validated benchmark dataset (Sec. 4.1) for spectrum-related Q&A. This is a valuable contribution that can foster future research.

**Weaknesses:**

1. The "Naive RAG" baseline is intentionally crippled by design (lacking SQL capability), resulting in a 0% win rate on structured data tasks. The claimed >85% "win rate" is thus an artifact, likely demonstrating the trivial conclusion that an SQL-capable system outperforms one that is not. These evaluation flaws make it impossible to validate the paper's central claims.

2. The "agentic" contribution appears to be overstated. The system architecture (Sec. 3) seems to be a simple router (Supervisor) and a monolithic "Compound Agent" (admitted in Sec. 5 as a workaround for communication issues), rather than a sophisticated multi-agent framework. The core complexities (SQL-gen, GraphRAG) are outsourced to SOTA foundation models (GPT-4/5). Furthermore, the method relies on an idealized assumption of perfect pre-processing from unstructured PDFs to structured databases (Sec. 3.3). This fragility is highlighted in the case study (Appx. C.2), where the proposed GraphRAG tool fails and must fall back to a basic RAG, undermining claims of robustness.

3. The reported high performance (Sec. 4.4) appears to be an artifact of the benchmark design rather than a genuine methodological advance. The resulting system lacks generalizability; it is a highly customized, brittle implementation that relies on significant offline pre-processing and is tightly coupled to a small, flawed dataset. The claims of "broader applications" (Abstract) are therefore not supported by the evidence provided.

**Questions:**

None

---

### Official Review · Reviewer_pDNB · 2025-11-01

**Soundness:** 3
**Presentation:** 3
**Contribution:** 3
**Rating:** 2
**Confidence:** 4

**Summary:**

The paper tackles the challenge of building a RAG system that can analyze large volumes of information from different sources such as spectrum policy making, which traditionally requires extensive manual review of regulatory documents and license records.
It proposes a multi-agent hybrid retrieval-augmented generation (RAG) framework that integrates SQL-based structured retrieval for license data and GraphRAG-based retrieval for unstructured policy documents, coordinated through specialized agents.
Experimental results on an expert-validated benchmark show that this system substantially outperforms standard and web-search RAG baselines in spectrum policy question answering

**Strengths:**

1. The problem is very well motivated, as spectrum policymaking naturally demands reasoning over large, heterogeneous, and regulation-heavy datasets, making it an excellent real-world use case for complex RAG systems.

2. The benchmark is useful, providing a real dataset specifically designed to evaluate complex, multi-step retrieval and analysis queries that go beyond the capabilities of standard RAG approaches.

**Weaknesses:**

The technical novelty is limited, the paper primarily integrates existing ideas from SQL-RAG, GraphRAG, and tool-augmented reasoning without introducing new learning algorithms or coordination strategies. The “multi-agent” structure functions as a rule-based router rather than a genuinely interactive or adaptive multi-agent system.

The benchmark dataset is small in scale, with only 140 QA pairs, which restricts the statistical strength of the evaluation and makes it difficult to generalize the reported good results.

The comparison set of baselines is weak, the experiments include only Naive RAG and Web-Search RAG, omitting stronger recent baselines such as Search-o1 or iterative agentic retrieval methods that perform multi-step reasoning.

**Questions:**

see weakness

---

### Note · Authors · 2025-11-19

I have read and agree with the venue's withdrawal policy on behalf of myself and my co-authors.